# Association between Endoscopic Milk-White Mucosa, Epithelial Intracellular Lipid Droplets, and Histological Grade of Superficial Non-Ampullary Duodenal Epithelial Tumors

**DOI:** 10.3390/diagnostics11050769

**Published:** 2021-04-25

**Authors:** Yuko Hara, Kenichi Goda, Shinichi Hirooka, Takehiro Mitsuishi, Masahiro Ikegami, Kazuki Sumiyama

**Affiliations:** 1Department of Endoscopy, The Jikei University School of Medicine, Tokyo 105-8461, Japan; kaz_sum@jikei.ac.jp; 2Gastrointestinal Endoscopy Center, Dokkyo Medical University, Tochigi 321-0293, Japan; goda@dokkyomed.ac.jp; 3Department of Pathology, The Jikei University School of Medicine, Tokyo 105-8461, Japan; shirooka@jikei.ac.jp (S.H.); ikegami@jikei.ac.jp (M.I.); 4Department of Pathology, Kawasaki Saiwai Hospital, Kanagawa 212-0014, Japan; takeccho86@gmail.com

**Keywords:** adenocarcinoma, adipose differentiation-related protein, duodenum, intraepithelial neoplasia, milk-white mucosa, superficial non-ampullary duodenal epithelial tumor

## Abstract

We previously reported that superficial non-ampullary duodenal tumors (SNADETs) commonly had a whitish mucosal surface, named milk-white mucosa (MWM). The aim of this study was to evaluate the association of MWM with epithelial intracellular lipid droplets (immunohistochemically stained by adipose differentiation-related protein (ADRP)) and histological tumor grades. We reviewed endoscopic images and the histopathology of SNADETs resected en bloc endoscopically. We analyzed the correlation between the positive rates of endoscopic MWM in preoperative endoscopy and resected specimens, and ADRP-positive rates in the resected specimens. Associations between the MWM-positive rates and tumor grades, high-grade intraepithelial neoplasia (HGIN)/intramucosal carcinoma (IC), and low-grade intraepithelial neoplasia (LGIN) were analyzed. All the 92 SNADETs analyzed were <20 mm and histologically classified into 39 HGIN/IC and 53 LGIN. Spearman’s rank correlation coefficient showed a significant correlation between MWM-positive and ADRP-positive rates (*p* < 0.001). MWM-positive rates were significantly lower in the HGIN/IC than in the LGIN in preoperative endoscopy (*p* < 0.001) and resected specimens (*p* = 0.02). Our results suggest that endoscopic MWM is closely associated with epithelial intracellular lipid droplets and that the MWM-positive rate may be a predictor of histological grade in small SNADETs.

## 1. Introduction

Duodenal adenocarcinoma, when diagnosed at advanced stages, is associated with a poor prognosis [1]. Pancreatoduodenectomy is the standard treatment for advanced duodenal adenocarcinoma in resectable cases; however, this procedure is extremely invasive, with a mortality rate substantially higher than that of other digestive tract surgeries, estimated at 5.2–8.5% [2,3,4,5]. In contrast, when found at an early or precancerous stage, superficial epithelial tumors can be resected curatively and in a minimally invasive manner. At non-ampullary sites of the duodenum, endoscopic submucosal dissection is recommended for superficial duodenal tumors with a diameter larger than 20 mm, as it is more likely to achieve en bloc resection [6]. However, endoscopic submucosal dissection is exceedingly difficult to perform in the duodenum and frequently causes complications such as intraoperative perforation, delayed perforation, and postoperative bleeding in 2–27%, 18%, and 9–18% of cases, respectively [7,8]. Thus, the early detection and characterization of duodenal epithelial tumors with a diameter smaller than 20 mm is important because the tumors can be resected en bloc safely using the conventional technique of endoscopic mucosal resection [8].

We previously proposed referring collectively adenomas (intraepithelial neoplasias) and superficial adenocarcinomas (confined to the submucosal layer) that occur in non-ampullary regions of the duodenum as superficial non-ampullary duodenal epithelial tumors (SNADETs) [8,9]. SNADETs of high-grade intraepithelial neoplasias (HGINs) [10] and intramucosal carcinoma (IC), which pose a negligible risk of lymph node metastasis, are well-indicated for endoscopic resection. The gold-standard method of preoperative diagnosis is histological diagnosis via biopsy. However, histology of the preoperative biopsy can sometimes be incorrect and differ from the final histology of resected specimens [9]. Furthermore, the mucosal defect during the biopsies exposes the submucosa to bile and pancreatic juices, which can cause severe inflammation and fibrosis that further complicate endoscopic resection [9].

Indeed, previous studies have demonstrated the poor accuracy of preoperative biopsy in making a differential diagnosis between low-grade intraepithelial neoplasia (LGIN) and HGIN/superficial adenocarcinoma, ranging from 67.6% [9] to 71.1% [11]. Therefore, biopsy-free endoscopic diagnostic methods are needed to predict the histological grade of SNADET lesions more accurately. 

SNADETs frequently exhibit a peculiar endoscopic finding that we first termed “milk-white mucosa (MWM),” which was pathologically suspected to correspond to the accumulation of lipid droplets inside neoplastic epithelial cells [12,13]. However, we could not advance our study on MWM and epithelial intracellular lipid droplets because the lipids are soluble in alcohol and lost by alcohol dehydration, which is the most commonly used technique for preparing tissue specimens for pathological diagnosis. Thus, in our previous study, we used frozen specimens for identifying epithelial intracellular lipid droplets [12]. However, we could not evaluate the association between MWM and histological grade of SNADET, because nuclear swelling/deformation is caused by the freezing process and the crushing of tissue structures during frozen sectioning, and these damages to the nuclei and structure precluded the histological diagnosis of the tumor grade. Hence, the association between the epithelial intracellular lipid droplets, the endoscopic MWM, and histology remains unclear. 

To overcome the limitations, in this study, we used an immunohistochemical technique of adipose differentiation-related protein (ADRP) staining. ADRP is associated with the formation of lipid droplets [14] and enables the confirmation of the presence of epithelial intracellular lipid droplets in alcohol-fixed paraffin-embedded specimens without the need for frozen sections [15]. Therefore, to evaluate the association between these two phenomena and explore their potential application in diagnosing histological grade of SNADET, we compared the endoscopic MWM findings to the immunohistochemistry of epithelial intracellular lipid droplets in resected specimens, using antibodies against ADRP. The aim of this study was to evaluate the association of the endoscopic MWM and epithelial intracellular lipid droplets by ADRP staining and to investigate the relationship between the endoscopic MWM-positive rate and histological grade of SNADET.

## 2. Materials and Methods

### 2.1. Samples

Among the 148 SNADET lesions in 133 patients that were endoscopically resected at our hospital between April 2004 and July 2017, we included SNADET lesions that were evaluated by narrow-band imaging magnification endoscopy (NBI-ME) and resected en bloc, resulting in 92 SNADET lesions in 89 patients available for one-to-one correspondence between endoscopic findings and histology. The exclusion criteria were as follows: lesions with piecemeal resection, out of focus NBI-ME images, and lesion surface crushed during endoscopic resection. All of the included SNADET lesions were sporadic tumors and there were no patients with inflammatory bowel disease, celiac disease, or familial adenomatous polyposis.

### 2.2. Endoscopic Evaluation

MWM was observed by NBI-ME more clearly than by standard white-light endoscopy [16,17]. Using NBI-ME images, endoscopic MWM-positive rates were determined as the rates of MWM presence in SNADET lesions in maximal sections of the resected specimens for evaluating the correlation between positive rates of endoscopic MWM and ADRP immunohistochemical staining. The definition of the maximal section was mentioned later (Section 2.3. Histopathological Evaluation). For evaluating the relationship between positive rates of endoscopic MWM and the histological grade of SNADET, an endoscopic MWM-positive rate was determined as the rate of MWM presence in the whole area of each SNADET lesion in the resected specimen and in preoperative endoscopy. Two GI endoscopists (Y.H. and K.G.) evaluated the MWM-positive rates and a final decision was established by a consensus between the two endoscopists who were blinded to the final histopathological diagnosis and immunohistochemistry of ADRP. In accordance with our previous study, an endoscopic MWM throughout or in a large part was rated as an “entire type,” whereas an endoscopic MWM only near the margin or no MWM was rated as a “non-entire type” [12]. Representative preoperative NBI-ME images of MWM-positive SNADETs are shown in Figure 1. 

### 2.3. Histopathological Evaluation

#### 2.3.1. Histological Grading of SNADETs

Histological grading was carried out using hematoxylin and eosin-stained resected specimens by two pathologists specializing in gastrointestinal cancer (S.H. and T.M.) blinded to the endoscopic findings. The histological grading was finally established by a consensus between the two pathologists. Three histological grades were assigned in accordance with the World Health Organization 2010 classification [10]: adenoma and adenocarcinoma, based on structural and cytological atypia [18]. Although the pathologists classified the resected adenoma lesions as LGIN or HGIN, it was impossible to clearly differentiate HGIN from IC. Thus, the two pathologists classified the resected SNADET lesions into two histological grades: LGIN and HGIN/IC. 

#### 2.3.2. Preparation for Evaluating Endoscopic MWM, Histology, and Immunohistochemistry

A GI endoscopist (K.G.) took photos of NBI-ME, including the whole area of each formalin-fixed resected specimen and cut the resected specimens with a 2- or 3-mm interval. We defined “the maximal section” as a cut specimen with the largest diameter including a central portion of each SNADET lesion. We then evaluated the endoscopic MWM-positive rates, both in the maximal section and in the whole area of SNADET in resected specimen. 

Thinly sliced sections of the cut specimens and the maximum section were prepared for hematoxylin–eosin (HE) staining and immunohistochemical staining for ARDP, respectively.

### 2.4. Immunohistochemical Procedure and Evaluation

Formalin-fixed and paraffin embedded tissue sections were prepared from endoscopically resected specimens. Immunohistochemistry was performed by autostainer (VENTANA Benchmark) with adipophilin antibody (AP125; 1:1000; Fitzgerald Industries International, Concord, MA, USA). I-VIEW™ DAB Universal Kit was used. For antigen retrieval, deparaffinized sections were heated at 100° for 30 min in a tris-based buffer, pH8.5 (Conditioning solution CC1 Ventana Medical System). The slides were treated with hydrogen peroxide (3% H_2_O_2_) for blocking endogenous enzymes and used for new vision blocking reagent. We evaluate the areas in the maximal tissue sections. The epithelial intracellular lipid droplets were observed as brownish, tiny granules (ADRP-positive areas). The two pathologists evaluated the ADRP-positive areas and their rates. As the positive control, we used normal sebaceous glands of the skin. 

Representative ADRP-positive and ADRP-negative SNADET lesions are shown in Figure 2. 

### 2.5. Statistical Analyses

We first evaluated the association between the distribution of MWM in endoscopic images (MWM-positive rates) and epithelial intracellular lipid droplets (ADRP-positive rates) on the surface of SNADETs, using Spearman’s rank correlation coefficient. Next, we compared the MWM-positive rates (0–100%) in resected specimens and in preoperative endoscopy between the LGIN and HGIN/IC groups.

Continuous variables are expressed as median values with interquartile ranges, and Mann–Whitney’s *U* test was performed to identify significant differences between two groups. The threshold of statistical significance was set at *p* < 0.05. All statistical analyses were carried out using Stata software version 14 (Stata Corp, College Station, TX, USA). 

## 3. Results

### 3.1. Clinicopathological Characteristics 

The second part of the duodenum was the most common tumor site (66%, 61/92), followed by the deeper duodenum (22%, third and fourth parts). The diameter of all tumors was less than 20 mm, and the median was 10 mm. Macroscopically, type 0–IIa lesions were the most common (45%, 41/92), followed by type 0–IIc. Histologically, the 92 SNADETs included 53 LGINs and 39 HGINs/ICs (Table 1).

### 3.2. Association between Endoscopic and Immunohistochemistry Findings

In the resected specimens, Spearman’s rank correlation coefficient showed a significant correlation between MWM-positive and ADRP-positive rates (correlation coefficient: 0.87, *p* < 0.001; Figure 3). In resected specimens, the MWM-positive rates were significantly lower in the HGIN/IC group (%, median (IQR): 20 (10–40)) than in the LGIN group (40 (20–70)) (Mann–Whitney *U* test, *p* = 0.02) (Figure 4A). In preoperative endoscopic images of SNADETs, the MWM-positive rates were significantly lower in the HGIN/IC group (30 (10–68)) than in the LGIN group (70 (35–90)) (Mann–Whitney U test, *p* < 0.001) (Figure 4B). The difference in MWM-positive rates between the LGIN and HGIN/IC groups was greater in the preoperative endoscopic findings than in the resected specimens. Spearman’s rank correlation coefficient showed a significant correlation between MWM-positive rates in the resected specimens and preoperative endoscopy (correlation coefficient: 0.48, *p* < 0.001).

## 4. Discussion

We identified a significant correlation between the endoscopic MWM-positive rate and immunohistochemical ADRP-positive rate in resected SNADET specimens, suggesting that epithelial intracellular lipid droplets can be endoscopically visualized as MWM. Furthermore, endoscopic MWM-positive rates correlated with the histological tumor grade and HGIN/IC showed significantly lower MWM-positive rates than LGIN. To the best of our knowledge, this is the first report of an immunohistochemically demonstrated association of endoscopic MWM findings, epithelial intracellular lipid droplets, and histological tumor grade.

In our previous study [12], we categorized 24 SNADET lesions into two types: those presenting endoscopic MWM throughout or in a large part (entire type) and those presenting MWMs only near the margins or no MWMs at all (non-entire type). We found that LGIN lesions are more likely to be the entire type, whereas HGIN/IC lesions are more likely to be the non-entire type, revealing a significant difference [12]. The present findings therefore support the results of our previous study.

Considering that endoscopic biopsy can result in severe fibrosis and make endoscopic resection more difficult in the duodenum than in the rest of the digestive tract, endoscopic grading using MWM findings (entire or non-entire type) may be useful for a biopsy-free diagnosis of HGIN/IC. As aforementioned, the early identification of HGIN/IC tumors smaller than 20 mm is important because the tumors can be resected en-bloc safely using the conventional technique of endoscopic mucosal resection. All SNADETs included in this study were smaller than 20 mm. Thus, these results suggest that a non-entire type MWM of small SNADETs (<20 mm) should be considered for endoscopic mucosal resection as a total biopsy to make a definitive histological diagnosis. 

Ueo et al. [19] noted the presence of white opaque substances (WOS) in gastric tumors, which are similar to our MWM in SNADETs, and used electron microscopy to study them in detail. They observed a multitude of small granular structures inside the cytoplasm of cells involved in WOS formation, which were identified as ADRP-positive lipid droplets [19]. Duodenal villi are known to whiten by absorbing lipids [20], and several hypotheses for the mechanism by which lipid droplets accumulate in neoplastic epithelial cells have been proposed. The “dysfunctional transport hypothesis” [19] (Figure 5) proposes that after being absorbed into the villous epithelium, lipids form chylomicrons and are excreted into the lymphatic ducts in the lamina propria. However, tumor formation may disrupt the function of basolateral membranes (green arrow in Figure 5) of epithelial cells, inhibiting lymphatic duct clearance and causing lipid droplets to accumulate inside epithelial cells. Additionally, if LGIN progresses to HGIN/IC, it may disrupt the function of not only the basolateral membranes (green arrow in Figure 5) but also the apical membranes (yellow arrow in Figure 5) and impair lipid absorption. This will decrease lipid droplet accumulation inside epithelial cells. Therefore, the accumulation of lipid droplets may indicate the occurrence of abnormal lipid metabolism caused by tumor formation and progression from LGIN to HGIN/IC in the duodenum.

Since we first suggested an association between MWM endoscopic findings and histological grade [12], Kikuchi et al. [16] reported that 67% of lesions with an extensive WOS distribution (i.e., entire-type MWM in our study) were LGINs. Kakushima et al. [17] also reported that all lesions with extensive WOS deposition similar to entire-type MWM were LGINs. This study is the first to use an immunohistochemical technique (ADRP staining) and validate the association between MWM and the histological grade of SNADETs. Additionally, we validated the results of our own and others’ previous studies. Non-entire type MWM may suggest HGIN/IC, which is a good indication for endoscopic mucosal resection because this study included only small SNADET lesions less than 20 mm.

Nevertheless, this study had some limitations. First, this was a single-center, retrospective study. Therefore, further research—ideally, a multicenter, prospective trial—is needed to validate these results in a large-scale study. Second, there was substantial selection bias in this study that could not be avoided, because only cases of SNADET with en bloc resection were included for one-to-one comparisons between endoscopic findings and histology. The substantial selection bias discouraged us from analyzing the accuracy values of endoscopic MWM. Third, the influence of histological artifacts caused by thermal coagulation during endoscopic resection could not be eliminated. Fourth, diagnostic reproducibility of MWM was not evaluated in this study. Further study is needed for interobserver variability in the endoscopic diagnosis of MWM.

## 5. Conclusions

In summary, we studied the association of the endoscopic MWM and epithelial intracellular lipid droplets by ADRP immunohistochemical staining, and investigated the relationship between the endoscopic MWM-positive rate and histological grade of SNADETs resected en bloc endoscopically.

Our results suggest that endoscopic MWM is closely associated with immunohistochemistry of epithelial intracellular lipid droplets, and that the degree of the endoscopic MWM-positive rate may be a predictive indicator of the histological grade of small SNADETs.

## Figures and Tables

**Figure 1 diagnostics-11-00769-f001:**
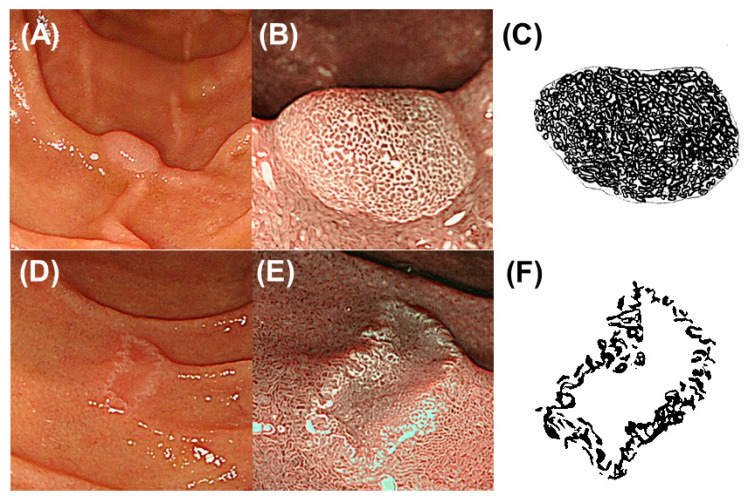
Representative images of entire-type and non-entire-type milk-white mucosa (MWM). (**A**,**B**) Representative endoscopic images of a superficial non-ampullary duodenal epithelial tumor (SNADET) with entire-type MWM. (**A**) White-light endoscopic image showing a small, elevated lesion. (**B**) Narrow band imaging (NBI) magnification endoscopic image showing MWM throughout the elevated lesion corresponding to entire-type MWM. (**C**) Schematic diagram of entire-type MWM. The black area shows the endoscopic MWM-positive area throughout the lesion. The endoscopic MWM-positive rate was 95%. (**D**,**E**) Representative endoscopic images of a SNADET with non-entire type MWM. (**D**) White-light endoscopic image showing a small, depressed lesion. (**E**) NBI magnification endoscopic image showing MWM only along or near the margin corresponding to non-entire type MWM. (**F**) Schematic diagram of non-entire type MWM. The endoscopic MWM-positive rate was 40%.

**Figure 2 diagnostics-11-00769-f002:**
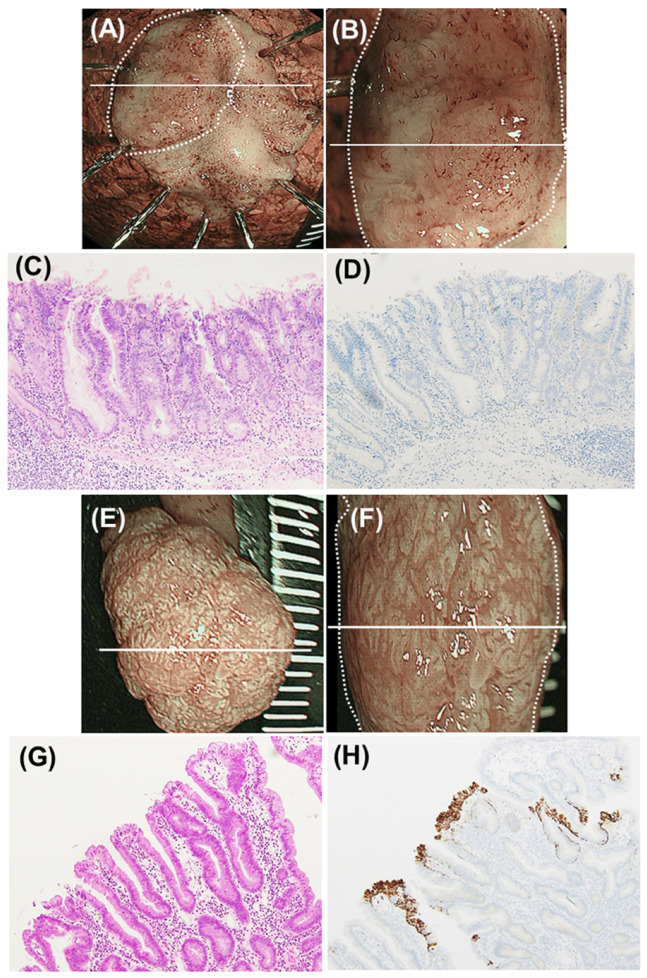
Representative images of milk-white mucosa (MWM)-positive and -negative superficial non-ampullary duodenal epithelial tumor (SNADET) lesions. (**A**–**D**) Representative images of a SNADET of high-grade intraepithelial neoplasia (HGIN). (**A**) Narrow band imaging (NBI) endoscopic image of an endoscopically resected specimen. The white line indicates the maximal section. The endoscopic MWM-positive rate was 0% (non-entire type). (**B**) NBI magnification endoscopic image of the resected specimen. The white line indicates the maximal section. (**C**) Histology of the maximal section (hematoxylin-eosin (HE) staining). (**D**) Immunohistochemical finding of the maximal section (adipose differentiation-related protein (ADRP) staining). Neoplastic epithelial cells were all negative for ADRP staining. (**E**–**H**) Representative images of a SNADET of low-grade intraepithelial neoplasia (LGIN). (**E**) NBI endoscopic image of an endoscopically resected specimen. The white line indicates the maximal section. The endoscopic MWM-positive rate was 95% (entire type). (**F**) NBI magnification endoscopic image of the resected specimen. The white line indicates the maximal section. (**G**) Histology of the maximal section (HE staining). (**H**) Immunohistochemical finding of the maximal section (ADRP staining). Most neoplastic epithelial cells (90%) were positive for ADRP staining.

**Figure 3 diagnostics-11-00769-f003:**
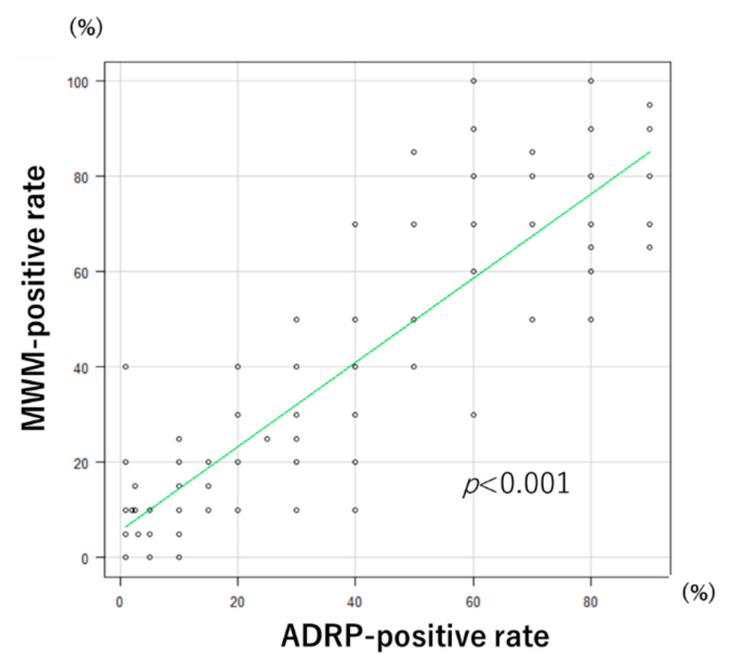
Correlation plots of endoscopic milk-white mucosa (MWM)-positive rates and adipose differentiation-related protein (ADRP)-positive rates.

**Figure 4 diagnostics-11-00769-f004:**
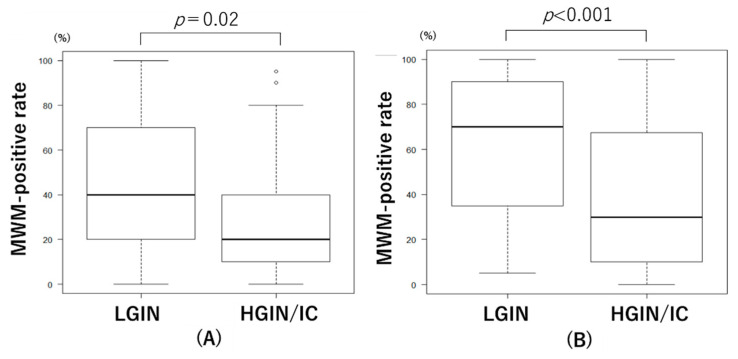
Relationship between histological tumor grades and endoscopic milk-white mucosa (MWM)-positive rates in resected specimens (**A**) and those in preoperative endoscopy (**B**). LGIN, low-grade intraepithelial neoplasia; HGIN, high-grade intraepithelial neoplasia; IC, intramucosal carcinoma.

**Figure 5 diagnostics-11-00769-f005:**
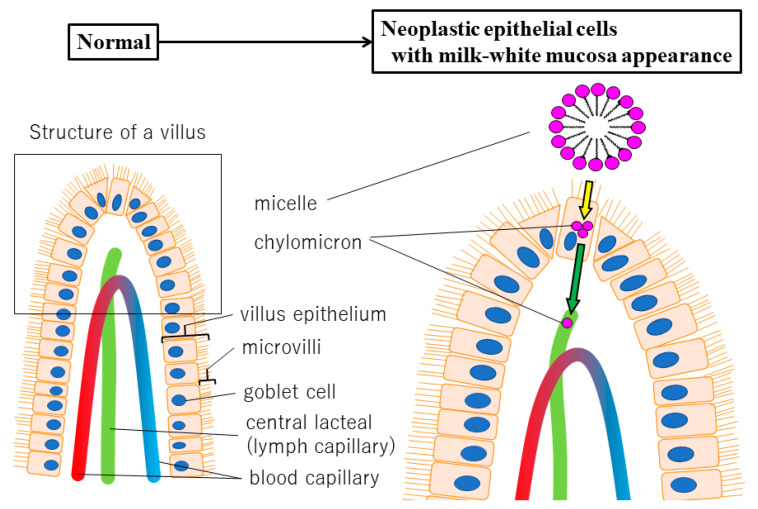
Duodenum normal villus histological structure and lipid accumulation based on the dysfunctional transport hypothesis.

**Table 1 diagnostics-11-00769-t001:** Patient and tumor characteristics (89 patients with 92 SNADETs).

Characteristic		Value
Sex (male), *n* (%)		67 (75)
Age (years), median (IQR)		65 (55–70)
Location in the duodenum (portion), *n*	First/Second/Third or Fourth	11/61/20
Diameter (mm), median (IQR)		10 (8–17)
Macroscopic type, *n*	0–I/IIa/IIc	18/41/33
Histological tumor grade, *n*	LGIN	53
	HGIN/IC	39

IQR, interquartile range; LGIN, low-grade intraepithelial neoplasia; HGIN, high-grade intraepithelial neoplasia; IC, intramucosal carcinoma.

## Data Availability

The data that support the findings of this study are available from the corresponding author upon reasonable request.

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
