# Peer review of "Association between Endoscopic Milk-White Mucosa, Epithelial Intracellular Lipid Droplets, and Histological Grade of Superficial Non-Ampullary Duodenal Epithelial Tumors"

_diagnostics, 2021, doi:10.3390/diagnostics11050769_

Round 1
Reviewer 1 Report
Hara et al. found a positive correlation between the protein level of ADRP and a whitish mucosal surface (found in endoscopy) in superficial non-ampullary duodenal tumors (SNADETs). They also found the association between this milk-white mucosa (MWM) with the histological grade in small SNADETs. Their finding suggests the MWM-positive rate may be used as a predictor in SNADETs. Though their findings were interesting, the effects of some factors were not examined in the manuscript. Several concerns are raised as listed below.
- The title is not very informative, authors need to modify their title.
- Authors found significant positive correlation between the ADRP protein level and MWM-positive rate. However, the MWM-positive rates were determined in NBI-ME images while the protein levels of ADRP were determined by immunohistochemical staining with tiny areas of samples. Given subregions from the same specimen may have variations in ADRP detection. Authors need to examine the variation of the protein expression levels of ADRP across the interested area.
- Following previous question, the maximal tissue sections were used to examine the association in this paper. The definition of “the maximal section” should be clearly addressed. Authors should make sure every endoscopist can define the same maximal section. For example, what if there are multiple areas having this kind of whitish mucosal surface?
- In figure 4, both (A) resected specimens and (B) preoperative endoscopy are from the same group of patients, authors may compare the MWM-positive rate between the same patient from resected specimens and preoperative endoscopy. This comparison can show how large the variation MWM-positive rates derived from endoscopy observation and endoscopically resected specimens are. MWM-positive rate can serve as a stable predictor only if the variation is low.
- The expression of ADRP has been found as a cancer-specific survival predictor in clear cell renal carcinoma (Yao et al., 2007). Given authors found the association between MWM, ADRP and histological grade, it is worthy examining whether MWM or ADRP protein expression level are predictors in SNADETs.
Author Response
<Reviewer 1>
Dear Reviewer
Thank you very much for reviewing our manuscript and offering valuable advice.
We have addressed your comments with point-by-point responses, and revised the manuscript accordingly.
Hara et al. found a positive correlation between the protein level of ADRP and a whitish mucosal surface (found in endoscopy) in superficial non-ampullary duodenal tumors (SNADETs). They also found the association between this milk-white mucosa (MWM) with the histological grade in small SNADETs. Their finding suggests the MWM-positive rate may be used as a predictor in SNADETs. Though their findings were interesting, the effects of some factors were not examined in the manuscript. Several concerns are raised as listed below.
- The title is not very informative, authors need to modify their title.
We would like to change the tile as mentioned below. We believe the revised title to be more informative and representing the aim and conclusion of this study.
Revised title: Association between endoscopic milk-white mucosa, epithelial intracellular lipid droplets, and histological grade of superficial non-ampullary duodenal epithelial tumors.
- Authors found significant positive correlation between the ADRP protein level and MWM-positive rate. However, the MWM-positive rates were determined in NBI-ME images while the protein levels of ADRP were determined by immunohistochemical staining with tiny areas of samples. Given subregions from the same specimen may have variations in ADRP detection. Authors need to examine the variation of the protein expression levels of ADRP across the interested area.
Response: We apologize for making you confused. The primary aim of this study is to verify whether epithelial intracellular lipid droplets is visualized as milk-white mucosa by endoscopy. The epithelial intracellular lipid droplets will show positive for ADRP staining in resected specimen. Hence, we compared positive rates of endoscopic milk-white mucosa and ADRP staining in the en-bloc resected specimen available for one-to-one correspondence between endoscopic findings and histology. We evaluate the endoscopic image of the resected specimen with cutting 2- or 3-mm interval (Actually, GI endoscopists cut resected specimen for histopathology in our hospital). For precise comparison with one-to-one correspondence, we focused on a section area of the maximum diameter and evaluate positive rates of endoscopic milk-white mucosa and ADRP in the maximal section. We revised and added sentences in the ‘Materials and Methods’ section as mentioned below.
(Materials and Methods)
2.2. Endoscopic evaluation
MWM was observed by NBI-ME more clearly than by standard white-light endoscopy [14,15]. Using NBI-ME images, endoscopic MWM-positive rates were determined as the rates of MWM presence in SNADET lesions in maximal sections of the resected specimens for evaluating the correlation between positive rates of endoscopic MWM and ADRP immunohistochemical staining. The definition of the maximal section was mentioned later (2.3.2). For evaluating the relationship between positive rates of endoscopic MWM and histological grade of SNADET, an endoscopic MWM-positive rate was determined as the rate of MWM presence in the whole area of each SNADET lesion in the resected specimen and in preoperative endoscopy. Two GI endoscopists (Y.H. and K.G.) evaluated the MWM-positive rates and a final decision was established by a consensus between the two endoscopists who were blinded to the final histopathological diagnosis and immunohistochemistry of ADRP. In accordance with our previous study, an endoscopic MWM throughout or in a large part was rated as an “entire type,” whereas an endoscopic MWM only near the margin or no MWM was rated as a “non-entire type” [12]. Representative preoperative NBI-ME images of MWM-positive SNADETs are shown in Figure 1.
2.3. Histopathological evaluation
2.3.1. Histological grading of SNADETs
Histological grading was carried out using hematoxylin and eosin-stained resected specimens by two pathologists specializing in gastrointestinal cancer (S.H. and T.M.) blinded to the endoscopic findings. The histological grading was finally established by a consensus between the two pathologists. Three histological grades were assigned in accordance with the World Health Organization 2010 classification [10]: adenoma and adenocarcinoma based on structural and cytological atypia [16]. Although the pathologists classified the resected adenoma lesions as LGIN or HGIN, it was impossible to clearly differentiate HGIN from IC. Thus, the two pathologists classified the resected SNADET lesions into two histological grades: LGIN and HGIN/IC.
2.3.2. Preparation for evaluating endoscopic MWM, histology, and immunohistochemistry
A GI endoscopist (K.G.) took photos of NBI-ME including the whole area of each formalin-fixed resected specimen and cut the resected specimens with 2- or 3-mm interval. We defined “the maximal section” as a cut specimen with the largest diameter including a central portion of each SNADET lesion. We then evaluated the endoscopic MWM-positive rates both in the maximal section and in the whole area of SNADET in resected specimen.
Thin sliced sections of the cut specimens and the maximum section were prepared for hematoxylin–eosin (HE) staining and immunohistochemical staining for ARDP, respectively.
- Following previous question, the maximal tissue sections were used to examine the association in this paper. The definition of “the maximal section” should be clearly addressed. Authors should make sure every endoscopist can define the same maximal section. For example, what if there are multiple areas having this kind of whitish mucosal surface?
Response: We defined “the maximal section” which was made from a cut specimen with the largest diameter only in the first half of this study for association between epithelial intracellular lipid droplets that will be positive for ADRP staining and endoscopic milk-white mucosa (MWM) by using resected specimens. We sought to evaluate the positive rates of endoscopic MWM and ADRP staining by using the tissue section made from a cut specimen with maximum diameter. In the second half of this study, we evaluated positive rates of MWM in whole lesion and association between these rates and histological grade. Thus, if results of this study applied to clinical endoscopy, endoscopists would need not to define the maximal section but to evaluate MWM in whole lesion for predicting histological grade. However, we could not evaluate interobserver variability in evaluating endoscopic MWM. I revised the manuscript in the ‘Materials and Methods’ and ‘Discussion’ sections mentioned below.
(Materials and Methods)
2.2. Endoscopic evaluation
MWM was observed by NBI-ME more clearly than by standard white-light endoscopy [14,15]. Using NBI-ME images, endoscopic MWM-positive rates were determined as the rates of MWM presence in SNADET lesions in maximal sections of the resected specimens for evaluating the correlation between positive rates of endoscopic MWM and ADRP immunohistochemical staining. The definition of the maximal section was mentioned later (2.3.2). For evaluating the relationship between positive rates of endoscopic MWM and histological grade of SNADET, an endoscopic MWM-positive rate was determined as the rate of MWM presence in the whole area of each SNADET lesion in the resected specimen and in preoperative endoscopy. Two GI endoscopists (Y.H. and K.G.) evaluated the MWM-positive rates and a final decision was established by a consensus between the two endoscopists who were blinded to the final histopathological diagnosis and immunohistochemistry of ADRP. In accordance with our previous study, an endoscopic MWM throughout or in a large part was rated as an “entire type,” whereas an endoscopic MWM only near the margin or no MWM was rated as a “non-entire type” [12]. Representative preoperative NBI-ME images of MWM-positive SNADETs are shown in Figure 1.
2.3. Histopathological evaluation
2.3.1. Histological grading of SNADETs
Histological grading was carried out using hematoxylin and eosin-stained resected specimens by two pathologists specializing in gastrointestinal cancer (S.H. and T.M.) blinded to the endoscopic findings. The histological grading was finally established by a consensus between the two pathologists. Three histological grades were assigned in accordance with the World Health Organization 2010 classification [10]: adenoma and adenocarcinoma based on structural and cytological atypia [16]. Although the pathologists classified the resected adenoma lesions as LGIN or HGIN, it was impossible to clearly differentiate HGIN from IC. Thus, the two pathologists classified the resected SNADET lesions into two histological grades: LGIN and HGIN/IC.
2.3.2. Preparation for evaluating endoscopic MWM, histology, and immunohistochemistry
A GI endoscopist (K.G.) took photos of NBI-ME including the whole area of each formalin-fixed resected specimen and cut the resected specimens with 2- or 3-mm interval. We defined “the maximal section” as a cut specimen with the largest diameter including a central portion of each SNADET lesion. We then evaluated the endoscopic MWM-positive rates both in the maximal section and in the whole area of SNADET in resected specimen.
Thin sliced sections of the cut specimens and the maximum section were prepared for hematoxylin–eosin (HE) staining and immunohistochemical staining for ARDP, respectively.
(Discussion)
Nevertheless, this study had some limitations. First, this was a single-center, retrospective study. Therefore, further research—ideally, a multicenter, prospective trial—is needed to validate these results in a large-scale study. Second, there was substantial selection bias in this study that could not be avoided because only cases of SNADET with en-bloc resection were included for one-to-one comparisons between endoscopic findings and histology. The substantial selection bias discouraged us from analyzing the accuracy values of endoscopic MWM. Third, the influence of histological artifacts caused by thermal coagulation during endoscopic resection could not be eliminated. Forth, diagnostic reproducibility of MWM was not evaluated in this study. Further study is needed for interobserver variability in the endoscopic diagnosis of MWM.
- In figure 4, both (A) resected specimens and (B) preoperative endoscopy are from the same group of patients, authors may compare the MWM-positive rate between the same patient from resected specimens and preoperative endoscopy. This comparison can show how large the variation MWM-positive rates derived from endoscopy observation and endoscopically resected specimens are. MWM-positive rate can serve as a stable predictor only if the variation is low.
Response: According to your suggestion, we added analysis for correlation coefficient between MWM-positive rates in preoperative endoscopy and resected specimens. The result showed a weak but a significant correlation between them (correlation coefficient: 0.48, p < 0.001). As mentioned in the “Results” section, although MWM-positive rates of HGIN/IC lesions were significantly lower than those of LGIN lesions both in preoperative endoscopy and resected specimens, the difference in MWM-positive rates between the LGIN and HGIN/IC groups was greater in the preoperative endoscopic findings than in the resected specimens. We believe these results suggest that MWM-positive rates may be a stable predictor and a better predictor for HGIN/IC in the preoperative endoscopy than in resected specimens. These will be favorable because we determine the management of SNADET cases in the preoperative endoscopy. We added sentences in the ‘Results’ section as mentioned below.
(Results)
3.2. Association between endoscopic and immunohistochemistry findings
In the resected specimens, Spearman’s rank correlation coefficient showed a significant correlation between MWM-positive and ADRP-positive rates (correlation coefficient: 0.87, p < 0.001; Figure 3). In resected specimens, the MWM-positive rates were significantly lower in the HGIN/IC group (%, median (IQR): 20 (10-40)) than in the LGIN group (40 (20-70)) (Mann–Whitney U test, p = 0.02) (Figure 4A). In preoperative endoscopic images of SNADETs, the MWM-positive rates were significantly lower in the HGIN/IC group (30 (10-68)) than in the LGIN group (70 (35-90)) (Mann–Whitney U test, p < 0.001) (Figure 4B). The difference in MWM-positive rates between the LGIN and HGIN/IC groups was greater in the preoperative endoscopic findings than in the resected specimens. Spearman’s rank correlation coefficient showed a significant correlation between MWM-positive rates in the resected specimens and preoperative endoscopy (correlation coefficient: 0.48, p < 0.001).
- The expression of ADRP has been found as a cancer-specific survival predictor in clear cell renal carcinoma (Yao et al., 2007). Given authors found the association between MWM, ADRP and histological grade, it is worthy examining whether MWM or ADRP protein expression level are predictors in SNADETs.
Response: Thank you very much for your valuable comments. As you indicated, expression level of ADRP was a cancer-specific survival predictor factor in clear cell renal carcinomas (Yao M, et al. Clin Cancer Res 2007). According to your suggestion, we evaluated positive rates of ADRP in the resected specimens of LGIN and HGIN/IC lesions. As the results, ADRP-positive rates were lower in HGIN/IC group (%, median (IQR): 20 (5-40)) than in the LGIN group (40 (10-70)), but the difference was not significant (Mann–Whitney U test, p = 0.069). The results may be derived from different nature between clear cell renal carcinomas and SNADETs.

Reviewer 2 Report
The manuscript is very interesting, dealing with an important topic, otherwise not well-explored. Therefore, it deserves to be published, but after minor revision that will include:
- extended Introduction providing more general information about the topic, which is commonly not well-known and explored
- information about patients involved and detailed immunohistochemistry procedure
- Discussion on the assumed composition of the intracellular lipid droplets and the onset of lipid peroxidation in respect to carcinogenesis (especially of stomach and gastrointestinal system in general), which might open options for complementary immunostainings, such as for lipid peroxidation products (like 4-hydroxynonenal protein adducts)
Author Response
<Reviewer 2>
Dear Reviewer
Thank you very much for reviewing our manuscript and offering valuable advice.
We have addressed your comments with point-by-point responses, and revised the manuscript accordingly.
The manuscript is very interesting, dealing with an important topic, otherwise not well-explored. Therefore, it deserves to be published, but after minor revision that will include:
extended Introduction providing more general information about the topic, which is commonly not well-known and explored
Response: Thank you for your nice advice. As mentioned below, we revised the ‘Introduction’ section including more information about the topics of this study including what is not well-known in previous studies and explored in this study.
(Introduction)
SNADETs frequently exhibit a peculiar endoscopic finding that we first termed as “milk-white mucosa (MWM),” which was pathologically suspected to correspond to the accumulation of lipid droplets inside neoplastic epithelial cells [12,13]. However, we could not advance our study on MWM and epithelial intracellular lipid droplets because the lipids are soluble in alcohol and lost by alcohol dehydration which is the most commonly used technique for preparing tissue specimens for pathological diagosis. Thus, in our previous study, we used frozen specimens for identifing epithelial intracellular lipid droplets [12]. However, we could not evaluate association between MWM and histological grade of SNADET because nuclear swelling/deformation caused by the freezing process and the crushing of tissue structures during frozen sectioning, and these damages to the nuclei and structure precluded histological diagnosis of tumor grade. Hence, the association between the epithelial intracellular lipid droplets, the endoscopic MWM, and histological remains unclear.
To overcome the limitations, in this study, we used an immunohistochemical technique of adipose differentiation-related protein (ADRP) staining. ADRP is associated with the formation of lipid droplets [14] and enables confirmation of the presence of epithelial intracellular lipid droplets in alcohol-fixed paraffin-embedded specimens without need for frozen sections [15]. Therefore, to evaluate the association between these two phenomena and explore their potential application in diagnosing histological grade of SNADET, we compared the endoscopic MWM findings to the immunohistochemistry of epithelial intracellular lipid droplets in resected specimens using antibodies against ADRP.
information about patients involved and detailed immunohistochemistry procedure
Response: According to your suggestion, we added sentences in the “Materials and Methods” as mentioned below.
(Materials and Methods)
2.1. Samples
Among the 148 SNADET lesions in 133 patients that were endoscopically resected at our hospital between April 2004 and July 2017, we included SNADET lesions that were evaluated by narrow-band imaging magnification endoscopy (NBI-ME) and resected en-bloc, resulting in 92 SNADET lesions in 89 patients available for one-to-one correspondence between endoscopic findings and histology. The exclusion criteria were as follows: lesions with piecemeal resection, out of focus NBI-ME images, and lesion surface crushed during endoscopic resection. All of the included SNADET lesions were sporadic tumors and there were no patients with inflammatory bowel disease, celiac disease, and familial adenomatous polyposis.
2.4. Immunohistochemical procedure and evaluation
Formalin-fixed and paraffin embedded tissue sections were prepared from endoscopically resected specimens. Immunohistochemistry was performed by autostainer (VENTANA Benchmark) with adipophilin antibody (AP125; 1:1000; Fitzgerald Industries International, Concord, MA, USA). I-VIEW™ DAB Universal Kit was used. For antigen retrieval, deparaffinized sections were heated at 100°for 30 minutes in a tris-based buffer, pH8.5 (Conditioning solution CC1 Ventana Medical System).The slides were treated with hydrogen peroxide (3% H2O2) for blocking endogenous enzymes and used for New Vision Blocking Reagent. We evaluate the areas in the maximal tissue sections. The epithelial intracellular lipid droplets were observed as brownish, tiny granules (ADRP-positive areas). The two pathologists evaluated the ADRP-positive areas and their rates. As the positive control, we used normal sebaceous glands of the skin. Representative ADRP-positive and ADRP-negative SNADET lesions are shown in Figure 2. We used the anti-adipophilin antibody (AP125; 1:1000; Fitzgerald Industries International, Concord, MA, USA); as the positive control, we used normal sebaceous glands of the skin.
Discussion on the assumed composition of the intracellular lipid droplets and the onset of lipid peroxidation in respect to carcinogenesis (especially of stomach and gastrointestinal system in general), which might open options for complementary immunostainings, such as for lipid peroxidation products (like 4-hydroxynonenal protein adducts)
Response: We appreciate for your interesting suggestion. As you indicated, studies showed relationships between lipid peroxidation, in particular 4-hydroxy-2-nonenal (HNE), within cells and the development of cancer including colorectal cancer (Guéraud F. Free Radic Biol Med. 2017; 111: 196-208.). As with colorectal cancer, HNE might be associated with the development of SNADETs, but we could not find research findings related to HNE and the development of duodenal cancer. Based on this study results, we deduced that the accumulation of epithelial intracellular lipid droplets was caused by development of SNADETs. However, the epithelial intracellular lipid droplets may not be a carcinogenic factor for duodenal cancer because endoscopic MWM which was corresponding to the epithelial intracellular lipid droplets significantly tended to disappear in cancerous lesions (HGIN/IC) of SNADETs. Although we can not understand lipid peroxidation (HNE) well and it will have a low potential for demonstrating an association, it may be an interesting study to explore the association between lipid peroxidation (HNE) and epithelial intracellular lipid droplets.
